# REASSESSING THE VALIDITY OF SPURIOUS CORRELATIONS BENCHMARKS

## ABSTRACT

Neural networks can fail when the data contains spurious correlations, i.e. associations in the training data that fail to generalize to new distributions. To understand this phenomenon, often referred to as subpopulation shift or shortcut learning, researchers have proposed numerous group-annotated spurious correlations benchmarks upon which to evaluate mitigation methods. However, we observe that these benchmarks exhibit substantial disagreement, with the best methods on one benchmark performing poorly on another. We explore this disagreement, and examine benchmark validity by defining three desiderata that a benchmark should satisfy in order to meaningfully evaluate methods. Our results have implications for both benchmarks and mitigations: we find that certain group-annotated benchmarks are not meaningful measures of method performance, and that several methods are not sufficiently robust for widespread use. We present a simple recipe for practitioners to choose methods using the *most similar* benchmark to their given problem.

## 1 INTRODUCTION

A striking failure mode of deep learning-based models is their susceptibility to spurious correlations, whereby models learn to use patterns that only hold in certain subsets of the data (Nagarajan et al., 2020; Geirhos et al., 2020). Researchers have produced numerous group-annotated benchmarks for evaluating and comparing methods for mitigating spurious correlations, ultimately informing decisions as to which method is best. In order to draw robust conclusions about which method to use, one would hope that different benchmarks produce similar results. However, in Figure 1 we observe this not to be the case: benchmarks often disagree, and methods that perform well on one benchmark perform poorly on others.

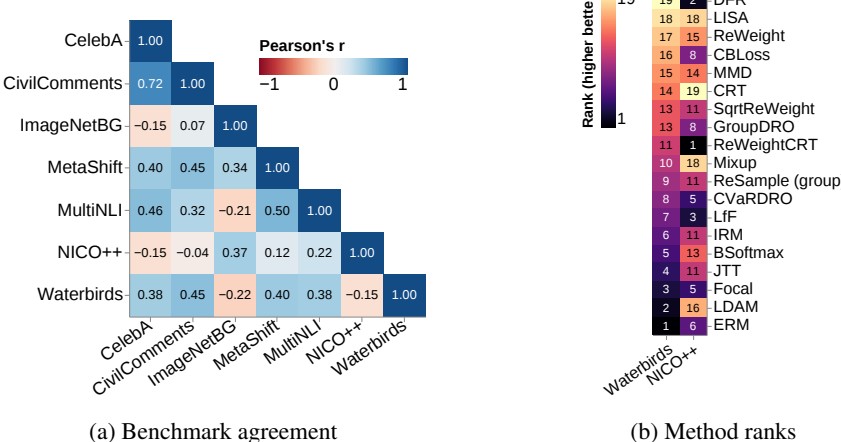

(a) Benchmark agreement

(b) Method ranks

Figure 1: **Spurious correlations benchmarks disagree.** **(a)** Correlation between worst-group accuracies on different benchmarks reported by Yang et al. (2023). **(b)** Waterbirds and NICO++ produce disagreeing ranks, such that the best method on Waterbirds (DFR) is the second worst on NICO++. Higher rank indicates stronger performance.

Faced with multiple benchmarks, standard machine learning practice is to average over contradictory results. Averaging, however, neglects that different benchmarks may measure different things, only some of which correspond to the desired quality. This presents a barrier to mitigating spurious correlations in practice. When confronted by a new dataset where a group attribute correlates with the target, which benchmark should one trust when deciding which method to apply?

In this work, we expose benchmark disagreement and analyze the validity of common group-annotated spurious correlations benchmarks. To do so, we suggest a set of properties that a spurious correlations benchmark should satisfy, and introduce a model-dependent statistic that quantifies the benchmark's task difficulty due to correlation between group attribute and target label. Using the established idea of convergent validity (Jacobs & Wallach, 2021), we expect that two valid benchmarks testing similar things—exhibiting similar task difficulty due to spurious correlation—should rank methods similarly.

**Our results reveal that certain group-annotated benchmarks are not valid tools for evaluating mitigation method performance.** Moreover, of all the methods evaluated here, only a small handful are robust to different tasks, while many exhibit strong performance only under specific conditions. Finally, we provide an approach to translating between benchmark results and real-world datasets, using our model-dependent statistic to understand which benchmark is most relevant.

## 1.1 BACKGROUND AND RELATED WORK

Models trained with empirical risk minimization (ERM; Vapnik, 1999, Ch. 1) tend to learn spurious correlations (Nagarajan et al., 2020), resulting in many real-world failures (Geirhos et al., 2020). For example, chest x-ray classifiers latch onto physical features of the scanner that fail to generalize to new hospitals (Zech et al., 2018). In hate speech detection, models use dialect differences rather than learning the desired task, resulting in a disproportionate false-positive rate (Sap et al., 2019). These spurious correlations often result in models which amplify bias (Zhao et al., 2017; Wang et al., 2019).

Many researchers have sought to understand why models rely on spurious correlations. Sagawa et al. (2020b) suggest that overparameterized models' bias against memorization leads them to rely on spurious correlations for minority samples. Given their well-known inductive bias towards simplicity (Kalimeris et al., 2019; Valle-Perez et al., 2018; Bell & Sagun, 2023), models use spurious correlations if they are simpler to learn than the intended function (Shah et al., 2020; Hall et al., 2022; Yang et al., 2024), where simplicity may be determined by model capacity (Sreekumar & Boddeti, 2023). Key factors determining the deleterious effect of a spurious correlation include the separability of the spurious features (Wang & Wang, 2024), correlation between the attribute and target (Yang et al., 2024; Deng et al., 2023), the relative signal-to-noise ratios of the core and spurious features (Yang et al., 2024), the number of spurious features Lin et al. (2023), and the relative size of the groups (Deng et al., 2023). Our effort to quantify task difficulty due to spurious correlation builds on these ideas of model-specific complexity, accounting for both correlation strength and learnability.

Yang et al. (2023) introduce *SubpopBench* and evaluate 22 mitigation methods over benchmarks exhibiting different types of *subpopulation shift*, including attribute and class imbalance and missing data. Our work builds on SubpopBench due to its comprehensive set of methods and benchmarks, though we narrow our focus to only spurious correlations. Several other benchmarking efforts exist, including those for evaluating spurious correlations mitigation (Joshi et al., 2023; Lynch et al., 2023), and other forms of subpopulation shift (Koh et al., 2021; Santurkar et al., 2020; Liu et al., 2023). More broadly, Gulrajani & Lopez-Paz (2020) introduce the *DomainBed* library for benchmarking performance in various domain generalization scenarios. Interestingly, both Gulrajani & Lopez-Paz and Joshi et al. find that with sufficient hyperparameter tuning ERM can be surprisingly robust, motivating our consideration of ERM failure as a necessary benchmark property.

Unlike previous work concerned with benchmarking existing mitigations methods, our aim is instead to investigate the validity of the benchmarks *themselves*. Our approaches are complementary: while new benchmarks and systematic evaluations are essential, so too are meta-analyses that help us make sense of conflicting results. Practically, we envisage our analysis supporting practical decisions around which benchmarks to rely upon, for example by filtering the benchmarks included in SubpopBench, or choosing between newer benchmark variants (e.g., Joshi et al., 2023; Lynch et al., 2023).

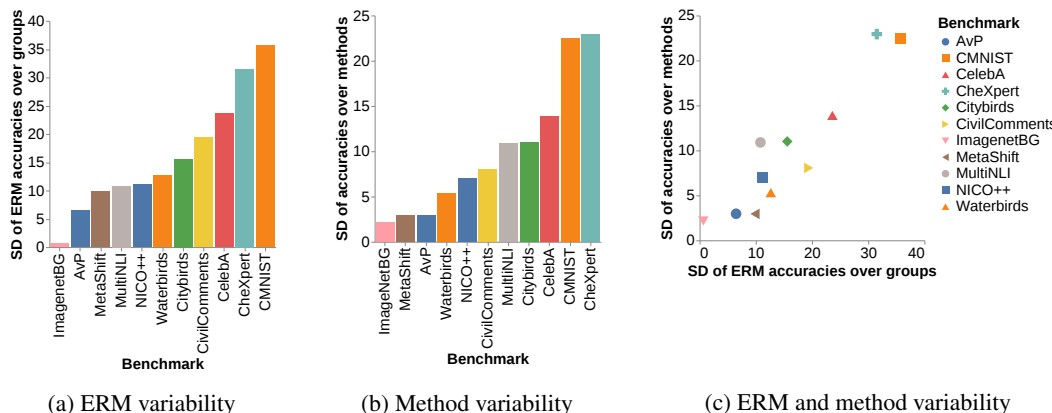

(a) ERM variability          (b) Method variability          (c) ERM and method variability

Figure 2: **(a)** Standard deviation (SD) of test accuracies over groups for an ERM-trained model. **(b)** SD of worst-group test accuracies over methods. **(c)** SD of ERM accuracies over groups vs. SD of worst-group test accuracies over methods. **Certain benchmarks, e.g. ImageNetBG, do not produce a "worst group", and result in tightly-clustered method performance.**

### 1.2 OUTLINE AND CONTRIBUTIONS

We begin in §2 with the simple observation that published results on spurious correlations benchmarks often disagree. This motivates our investigation of *what* benchmarks are measuring, and whether some benchmarks are more valid than others.

In §3, we introduce three desiderata—*ERM Failure*, *Discriminative Power*, and *Convergent Validity*—that capture key properties that a spurious correlations benchmark should satisfy in order to be a meaningful test of mitigation performance. Our test for Convergent Validity will call for some way of understanding precisely what a benchmark is actually measuring. To address this, in §3.4.1 we propose a model-dependent statistic that measures the difficulty of a task due to spurious correlation.

In §4 we analyze the validity of eight benchmarks and the robustness of 22 mitigation methods. Finally, in §5 we present a recipe for mapping between benchmarks and real-world datasets, and evaluate our approach on geographically-diverse image classification. We conclude with the broader implications of our work in §6.

### 2 NOT ALL BENCHMARKS AGREE

We define a benchmark as a pair of a task dataset (e.g. an image classification task) and an evaluation metric for ranking methods (e.g. worst-group test accuracy). A spurious correlations benchmark is designed to measure how well methods can *mitigate* the effects of spurious correlations. These are typically designed so that conventional training yields poor performance on certain subsets of the data, referred to as *groups*. If a benchmark's function is to allow us to conclude which methods are best, we would ideally like different benchmarks to agree with one another.

To our surprise, we find that they do not. Figure 1a shows the (dis)agreement, as measured by Pearson's $r$, in worst-group test accuracies over benchmarks reported by Yang et al. (2023) (see Appendix A). Two popular benchmarks, Waterbirds and CelebA, produce only mildly-correlated results, while results on Waterbirds and NICO++ are negatively correlated. This has a practical outcome: Figure 1b shows that **the best** performing method on Waterbirds is **the second-worst** on NICO++. This disagreement is not due to benchmark saturation, as Waterbirds and NICO++ worst-group test accuracies have different means ($\bar{x} = 78.4$ and $37.8$ respectively) but similarly large standard deviations ($s = 6.2$ and $5.8$). We report similar findings on WILDS (Koh et al., 2021) (Appendix A). This poses a challenge for the practitioner looking to mitigate spurious correlations: faced with inconsistent benchmarks, which method to use?

We propose a simple methodology to aid this problem by studying why benchmarks disagree, and evaluate whether some benchmarks are more valid measures of mitigation performance than others. To do so, we define three desiderata that valid spurious correlations benchmarks should respect.

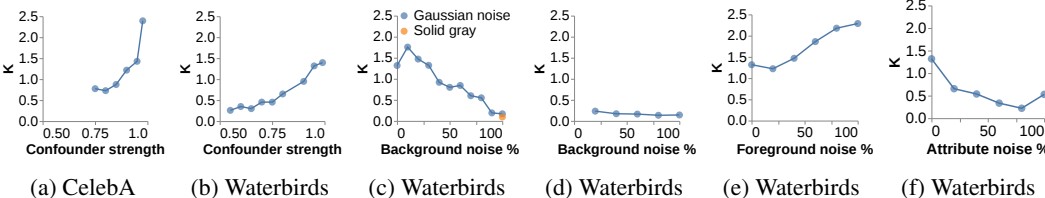

| (a) CelebA | (b) Waterbirds | (c) Waterbirds | (d) Waterbirds | (e) Waterbirds | (f) Waterbirds |

Figure 3: Task difficulty due to spurious correlation, as measured by Bayes Factor $K$, on modified benchmarks. Increasing the label-attribute correlation **(a, b)** and foreground noise **(e)** increases $K$, while increasing background noise **(c)** or applying a solid gray background (**c**, orange point) decreases $K$, except in the case where there is no correlation **(d)**. Attribute noise degrades the efficacy of $K$ **(f)**.

## 3   NOT ALL BENCHMARKS ARE VALID

For a benchmark to be a meaningful test of the ability to mitigate spurious correlations, we suggest it should satisfy three desiderata: *ERM Failure*, *Discriminative Power*, and *Convergent Validity*.

**ERM Failure** (§3.2). Spurious correlations mitigation methods are intended to prevent models from learning patterns that might perform well on average, but cause failures for certain groups. Thus, any benchmark intended to evaluate these methods should induce this problem when training with conventional empirical risk minimization (ERM), which by definition optimizes to minimize error averaged over all samples. **To satisfy ERM Failure, a benchmark should produce between-group performance disparities for models trained with ERM.**

**Discriminative Power** (§3.3). All benchmarks are intended for evaluation and comparison, and to support reasoning about *which* methods are best. In order to serve this purpose, a benchmark must discriminate between methods and assign different scores to each. These scores, in the spurious correlations setting, are typically worst-group test accuracies (Sagawa et al., 2020a). **To satisfy Discriminative Power, a benchmark should produce different worst-group test accuracies for different methods.**

**Convergent Validity** (§3.4). Even if a benchmark satisfies both ERM Failure and Discriminative Power, it still needs to rank methods *in a meaningful way*. In other words, we want our benchmark to exhibit *construct validity* (Jacobs & Wallach, 2021; Blodgett et al., 2021), i.e. it should allow us to truly measure the extent to which methods mitigate spurious correlations. Establishing construct validity of a benchmark is challenging without a ground truth, though one approach is to consider how the benchmark performs in relation to other benchmarks (that are themselves valid according to our first two desiderata). **To satisfy Convergent Validity, a benchmark should agree with other** *similar* **benchmarks, and disagree with those that are** *dissimilar*.

### 3.1   EVALUATION SETUP AND BENCHMARKS

We evaluate the validity of eight benchmarks included in SubpopBench: *Waterbirds* (Sagawa et al., 2020a), *CelebA* (Liu et al., 2015), *ImageNetBG* (a.k.a. *IN-9L Original*; Xiao et al., 2021), *MetaShift* (Indoor/Outdoor Cat vs. Dog; Liang & Zou, 2022), *NICO++* (Zhang et al., 2023), *CheXpert* (Irvin et al., 2019), *CivilComments* (Borkan et al., 2019), and *MultiNLI* (Williams et al., 2018).

In addition, we develop two new benchmarks to sanity check our approach. *Citybirds* is a clone of Waterbirds where confounding backgrounds are replaced with urban or rural scenes, such that Waterbirds and Citybirds should be equally valid. *Animals vs. Plants (AvP)* is a binary image classification task of animals and plants from Asia and Europe. Geography is difficult to infer relative to class membership,[1] so AvP has no practical spurious correlation. Finally, as a real-world test case in §5, we additionally evaluate on Dollar Street (Gaviria Rojas et al., 2022), a multiclass classification task over household objects, where groups are geographic regions. See Appendix D for full details.

We follow Yang et al.'s methodology (see Appendix C), report worst-group test accuracies, and perform model selection according to worst-group validation accuracies.

---

[1]A linear classifier over pretrained representations failed to achieve better than chance performance on region identification, versus strong performance on the task. See Appendix D.

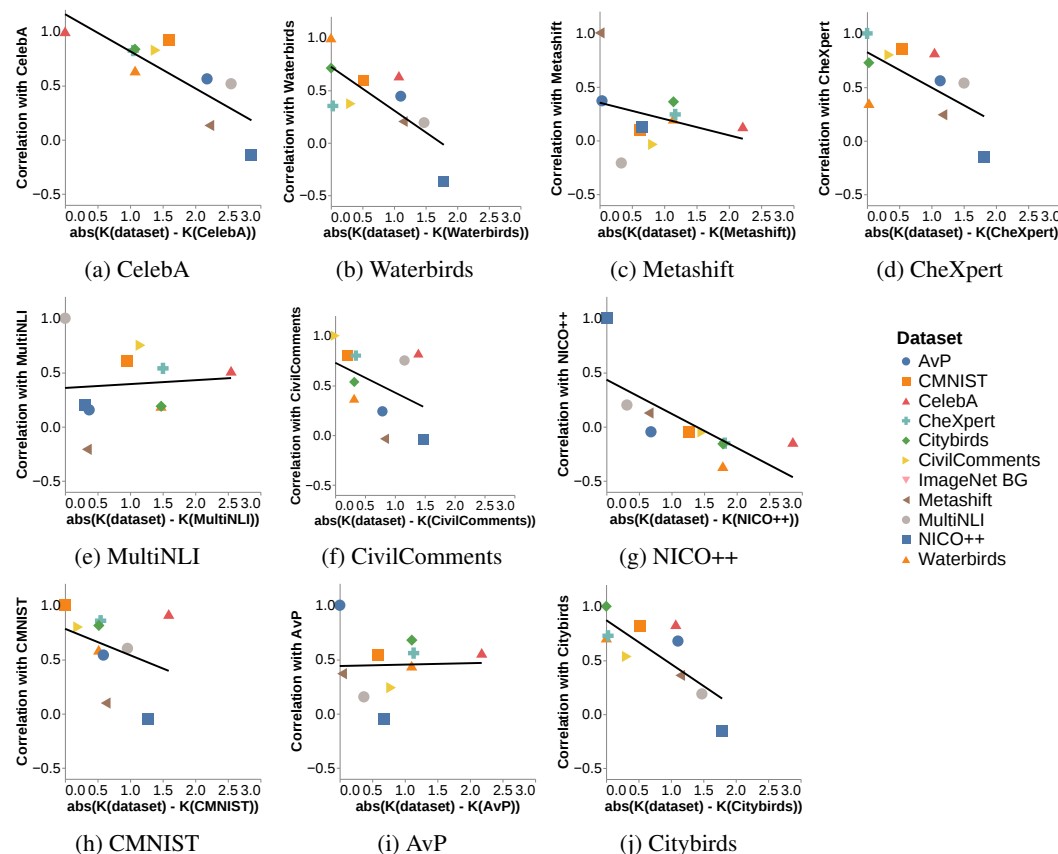

Figure 4: Benchmark agreement (Pearson's $r$) as a function of difference in task difficulty due to spurious correlation, as measured by Bayes Factor $K$. Each panel shows the agreement in worst-group test accuracies on the named dataset vs. all other datasets. Only benchmarks valid according to ERM group variability and method variability are included. **Valid benchmarks should agree more strongly with those that exhibit a similar $K$, thus exhibiting a negative correlation.** Black solid line fit with OLS linear regression.

## 3.2 EVALUATING ERM FAILURE

We test for ERM Failure by evaluating the variability of per-group test accuracies of ERM-trained models. High variability indicates that certain groups have worse performance than others, whereas low variability indicates that there is no real "worst group", thus not satisfying the ERM Failure desideratum.

Figure 2a shows the standard deviation (SD) of per-group mean test accuracies for an ERM-trained model. Immediately, we notice that ImageNetBG exhibits very low between-group variability and thus does not satisfy ERM Failure.

## 3.3 EVALUATING DISCRIMINATIVE POWER

To evaluate Discriminative Power, we test whether benchmarks assign different scores to different methods, by measuring the SD over method worst-group test accuracies for each dataset. Low SD indicates that all methods perform similarly, so the benchmark cannot discriminate between them, therefore not satisfying Discriminative Power.[2]

---

[2]A limitation of our approach is that low Discriminative Power could also be the result of all methods exhibiting similarly poor performance, such that the benchmark is valid but all methods are insufficiently powerful. Given the inclusion of ERM in the set of methods, and assuming ERM Failure is satisfied, we don't imagine this to be of practical concern.

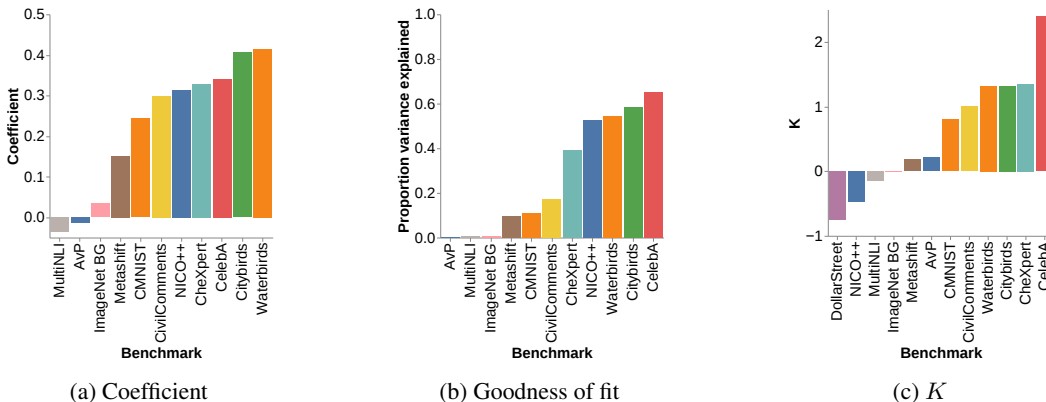

(a) Coefficient  (b) Goodness of fit  (c) $K$

Figure 5: **(a)** Benchmark agreement (Pearson's $r$) as a function of by difference in task difficulty due to spurious correlation ($K$). I.e., the (negative) slope of each line in Figure 4. **(b)** $R^2$ of each line. **(c)** Task difficulty due to spurious correlation, $K$. **Valid benchmarks should most agree with other benchmarks with similar $K$, so a large coefficient indicates a more valid benchmark.**

Figure 2b shows the SD over methods in SubpopBench. ImageNetBG results in tightly-clustered worst-group accuracies, not satisfying Discriminative Power. We also observe a strong positive correlation between ERM accuracy variability across groups and worst-group test accuracy variability across methods (Figure 2c). We hypothesize that benchmarks *could* exist that satisfy ERM Failure but not Discriminative Power, but we do not observe them here.

### 3.4 EVALUATING CONVERGENT VALIDITY

We suggest that the defining characteristic of a spurious correlations benchmark is the *task difficulty due to spurious correlation*. Thus, to test Convergent Validity we check if two benchmarks with similar task difficulty produce similar results. Specifically, we measure how inter-benchmark agreement changes as a function of the difference in task difficulty due to spurious correlation. Datasets that exhibit convergent validity should show a strong correlation, where increasing distance increases disagreement.

### 3.4.1 INTRODUCING $K$: TASK DIFFICULTY DUE TO SPURIOUS CORRELATION

Before we can evaluate Convergent Validity, we must first take a brief detour to understand what benchmarks actually measure. We argue that three factors should determine how a spurious correlations benchmark behaves:

1. The strength of the association, i.e. how often targets and attributes co-occur in the data;
2. The difficulty of learning the correlated attribute, i.e. how easily can a model predict the attribute; and
3. The difficulty of learning the intended target, i.e. how easily the model can predict the class label.

A common measure of spurious correlation is mutual information (MI) between groups and targets (Yang et al., 2023). However, MI only accounts for factor 1, and is not sensitive to factors 2 and 3, which are necessarily model-dependent.[3] See Appendix B for empirical evidence of this problem. To account for all three factors, we propose using the Bayes Factor as a model-dependent statistic that quantifies the task difficulty due to spurious correlations. The Bayes Factor evaluates the relative model performance of a model that *can* leverage the spurious correlation to solve the task, and a model that is penalized for doing so.

---

[3]Consider a grayscale-only model trained on Coloured MNIST. While there may be a spurious correlation between target and attribute, the model cannot exploit it (Arjovsky et al., 2019).

Table 1: Measures of validity, ERM Failure, Discriminative Power, and Convergent Validity, alongside best performing method and resulting worst-group test accuracies, for nine benchmarks from SubpopBench and our two additions. **Only certain benchmarks satisfy all three desiderata.**

| Benchmark | K | ERM Fail. | Disc. Power | Conv. Validity | Method | WG test acc. |
|---|---|---|---|---|---|---|
| MultiNLI | -0.15 | 10.81 | 10.89 | -0.04 | GroupDRO | 75.58 |
| AvP | 0.22 | 6.47 | 2.96 | -0.01 | GroupDRO | 91.75 |
| ImageNetBG | -0.01 | 0.65 | 2.16 | 0.03 | CRT | 78.22 |
| Metashift | 0.18 | 9.84 | 2.95 | 0.15 | ReSample | 80.00 |
| CMNIST | 0.80 | 35.68 | 22.45 | 0.24 | LISA | 71.09 |
| CivilComments | 1.00 | 19.40 | 8.06 | 0.30 | GroupDRO | 72.53 |
| NICO++ | -0.47 | 11.14 | 7.00 | 0.31 | Focal | 37.78 |
| CheXpert | 1.35 | 31.52 | 22.94 | 0.33 | CBLoss | 75.08 |
| CelebA | 2.39 | 23.61 | 13.90 | 0.34 | DFR | 87.78 |
| Citybirds | 1.32 | 15.57 | 11.01 | 0.41 | ReWeight | 90.50 |
| Waterbirds | 1.32 | 12.65 | 5.31 | 0.41 | LISA | 86.98 |

We use the Bayes Factor to compare how well the original and penalized models explain the data, defining the task difficulty due to spurious correlation of a benchmark as

$$K = \log \frac{P(Y_{\text{test}}^{\text{WG}}|X_{\text{test}}^{\text{WG}}, M_{RW})}{P(Y_{\text{test}}^{\text{WG}}|X_{\text{test}}^{\text{WG}}, M_{ERM})} \,,$$

where the numerator is the likelihood of the worst test group $(Y_{\text{test}}^{\text{WG}}, X_{\text{test}}^{\text{WG}})$, according to the model penalized for using the spurious correlation, $M_{RW}$, and the denominator is according to the model that uses the spurious correlation, $M_{ERM}$. $M_{ERM}$ is the benchmark's base model trained using ERM, whereas $M_{RW}$ is the same base model trained with a reweighted loss function, where each sample's weight is inversely proportional to its group size. We choose reweighting for its simplicity and minimal assumptions. In Appendix F we present almost identical results using an implementation of $K$ with GroupDRO, additionally finding $K$ is highly robust to hyperparameter tuning for both $M_{ERM}$ and $M_{RW}$. We use ResNet-50 (He et al., 2016) as a base model for vision benchmarks, and BERT for language (Devlin et al., 2019).

One can interpret $K$ as measuring how much better the reweighted model explains the test set, versus the ERM model. High $K$ indicates that the task is made more difficult by the spurious correlation, and a low $K$ indicates that the spurious correlation is not a dominant factor.

### 3.4.2 SANITY CHECKING $K$

To ensure that $K$ functions correctly, we test it using artificially manipulated datasets. We produce versions of CelebA and Waterbirds with increasingly correlated attributes and targets ("confounder strength"), and versions of Waterbirds with various amounts of background noise, foreground noise and attribute noise (see Appendix E). Figure 3 shows that, as intended, $K$ increases with confounder strength on both Waterbirds and CelebA (Figures 3a and 3b). On Waterbirds, $K$ decreases as background noise increases (reducing the utility of the spurious correlation), as long as a correlation exists (Figures 3c and 3d). Conversely, $K$ increases as foreground noise increases (increasing the utility of the spurious correlation; Figure 3f). We also see that Citybirds has a $K$ equal to that of Waterbirds, and AvP has low $K$ reflecting the limited utility of the spuriously correlated geographic information (Figure 5c). Our experiments support $K$ as a measure of task difficulty due to spurious correlation, though we note that $K$ depends on both the availability and quality of attribute annotations (Figure 3f).[4]

### 3.4.3 MEASURING CONVERGENT VALIDITY WITH $K$

Figure 5c shows the value of $K$ for our sample of benchmark datasets. Certain benchmarks, e.g. CelebA, have very high $K$, indicating that task difficulty is dominated by the spurious correlations in the data. Other datasets, such as Dollar Street and NICO++, exhibit low $K$, suggesting spurious correlation is not a principal factor in task difficulty.

---

[4]We consider 100% attribute noise equivalent to attribute information not being available.

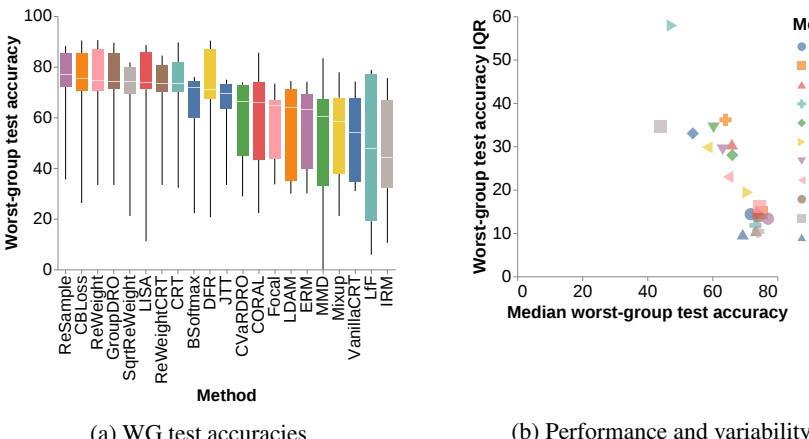

(a) WG test accuracies                  (b) Performance and variability

Figure 6: **(a)** Box plot of worst-group test accuracies for each method over all datasets. White line median; box IQR; whiskers range. **(b)** Median and IQR of worst-group test accuracies for each method. **Only certain methods consistently yield high worst-group accuracies.**

Recall in §3.4 we said that benchmarks exhibiting Convergent Validity must agree with other benchmarks that measure the same thing. More precisely, if we use $K$ to describe what a benchmark measures, valid benchmarks should agree most strongly with other datasets with a similar task $K$. In other words, between-benchmark disagreement should increase as two datasets have more different values of $K$. Figure 4 shows the the agreement (Pearson's $r$) in worst-group test accuracies achieved over methods between pairs of benchmarks, as a function of the distance in $K$ between the benchmarks. A strong negative slope indicates that the more benchmarks differ, the more they disagree, whereas a horizontal slope indicates that agreement is not a function of $K$ difference, i.e. not satisfying convergent validity. Figure 5a shows the (negative) slope of the agreement function for each benchmark. MultiNLI, AvP and ImageNetBG have low Convergent Validity.

**Summary.** ImageNetBG satisfies none of our three desiderata, whereas MultiNLI and AvP satisfy ERM Failure and Discriminative Power but not Convergent Validity. See Table 1.

## 4   NOT ALL METHODS ARE ROBUST

The function of a benchmark is to evaluate method efficacy to support reasoning about which method to use. Having discarded certain benchmarks as invalid measures, we now ask whether some methods are more robust than others. As a step towards making practical recommendations in real-world contexts, we ask whether certain methods are more robust to benchmarks with different $K$.

Figure 6a shows the distribution of worst-group test accuracies over benchmarks for each method, while Figure 6b shows the median worst-group test accuracy and the inter-quartile range (IQR).[5] Methods in the lower-right corner of Figure 6b, such as CRT, ReWeight, ReSample and GroupDRO, exhibit both high performance and low variability over benchmarks.

Next, we ask to what extent method variability is a function of the benchmark's task difficulty due to spurious correlation. Figure 7 shows how much a benchmark's $K$ explains the variance in worst-group test accuracies (see Appendix G). Many of the methods with low IQRs have a large proportion of their

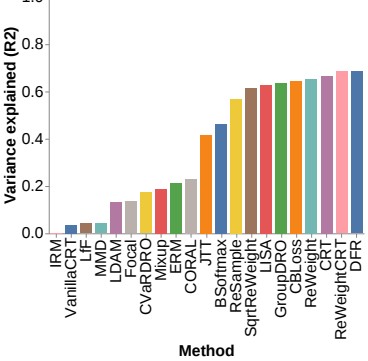

Figure 7: Proportion of worst-group test accuracy variance explained ($R^2$) by $K$ for each method over all datasets. Benchmark $K$ explains a large proportion of variance for stable methods.

---

[5]We use median and IQR as many methods exhibit non-normal performance distributions over benchmarks.

Table 2: Results of selecting a method according to worst-group test accuracy averaged over all benchmarks ("All Benchmarks"), averaged over valid benchmarks ("Valid Benchmarks"), and on the closest valid benchmark according to $K$ ("Closest Benchmark"). **Using the closest benchmark (col. 8) results in the best performance for 5 out of 8 test datasets**[7]. Alternatively, averaging over valid benchmarks (col. 5) instead of all benchmarks (col. 3) improves performance on 6 out of 9 test datasets. **On Dollar Street, using the closest benchmark improves on standard practice.**

| | All Benchmarks | | Valid Benchmarks | | Closest Benchmark | | |
| --- | --- | --- | --- | --- | --- | --- | --- |
| Test Dataset | Method | Acc. | Method | Acc. | Closest benchmark | Method | Acc. |
| CelebA | GroupDRO | 87.22 | ReSample | 85.37 | CheXpert | CBLoss | **87.41** |
| CheXpert | GroupDRO | 71.82 | ReSample | **74.21** | Citybirds | ReWeight | 73.95 |
| Citybirds | GroupDRO | **89.46** | ReSample | 88.16 | Waterbirds | LISA | 88.52 |
| CivilComments | GroupDRO | **72.53** | ReSample | 72.50 | - | - | - |
| CMNIST | GroupDRO | 70.83 | ReSample | 70.99 | Waterbirds | LISA | **71.09** |
| Metashift | GroupDRO | 75.90 | ReSample | **80.00** | CMNIST | LISA | 72.31 |
| NICO++ | GroupDRO | 33.33 | ReSample | **35.56** | Metashift | ReSample | **35.56** |
| Waterbirds | GroupDRO | 84.79 | ReSample | 85.57 | Citybirds | ReWeight | **86.68** |
| DollarStreet | GroupDRO | 58.65 | ReSample | 58.81 | NICO++ | Focal | **59.25** |

variance explained by the benchmark's $K$, suggesting that while variability may be reduced, they remain sensitive to benchmark specifics.

**Summary.** While practitioners must consider the specific nature of their dataset, certain methods, e.g. CRT, ReWeight, ReSample and GroupDRO, achieve consistently high worst-group test accuracies.

## 5 PRACTICAL RECOMMENDATIONS

When competing benchmarks suggest different methods, practitioners are left with a crucial question: what should I use for *my* specific dataset? Building upon our investigation of benchmark validity, we take a first step towards bridging the gap between real-world problems and common benchmarks.

A typical strategy is to take the method with the best performance averaged over all benchmarks. We evaluate two improvements over this conventional practice. First, we suggest only averaging over benchmarks deemed valid according to our three desiderata. Second, we suggest choosing the most similar benchmark to our given problem. Concretely, given some arbitrary dataset (which we refer as a "test dataset"[6]) we recommend that practitioners first calculate its $K$, and select the benchmark with the closest $K$. We hypothesize that the best performing methods on the closest benchmark will be most appropriate for the test dataset.

We test our approach using a Leave-One-Out analysis, where for each of eight test datasets we select the next closest benchmark, and evaluate the closest benchmark's best performing method on our test dataset ("Closest Benchmark"). We compare worst-group test accuracies for this method against the best method according to the all-benchmark average ("All Benchmarks"), and the valid-benchmark average ("Valid Benchmarks"). We perform an additional evaluation using Dollar Street, a dataset of geographically-diverse household images. Previous work has identified significantly worse performance of contemporary vision models for non-Western regions (de Vries et al., 2019; Richards et al., 2023). We extend SubpopBench to include a 42-class object classification task using Dollar Street, where group information is geographic region. See Appendix D for further details.

Table 2 shows the results of our two approaches. Averaging over benchmarks results in a single "best" method. Over all benchmarks, this method is GroupDRO; over valid benchmarks only this is ReSample. For six of nine test datasets, ReSample outperforms GroupDRO, confirming that filtering benchmarks before averaging is helpful. For five of eight test datasets,[7] selecting a method according to the closest benchmark would produce superior worst-group test accuracies, supporting our hypothesis that selecting based on similarity improves performance.

---

[6]We describe these as test datasets, rather than benchmarks, to emphasize we are evaluating our approach *as if we were a practitioner*, faced with some new test dataset.

[7]We exclude CivilComments as it lacks appropriate comparison benchmarks. Recall that we found MultiNLI to be invalid, and that $K$ is model-specific, such one can't compare across a ResNet-based and a BERT-based $K$.

On Dollar Street, selecting a method by averaging over valid benchmarks improves over indiscriminate averaging, and selecting a method using the closest benchmark improves performance further. We highlight that in Figure 5c we see a low $K$ for Dollar Street, which suggests that it is *not* a dataset dominated by spurious correlations, and other factors may contribute (Gustafson et al., 2023).

**Summary.** Practitioners should only average over valid benchmarks when determining which method is best. Choosing methods according to the closest benchmark may improve performance.

## 6 DISCUSSION

**Benchmark validity.** Our work joins a wider discussion evaluation practice validity, such as that of Jacobs & Wallach (2021) who argue that machine learning researchers can "collapse the distinctions between constructs and their operationalizations" (p. 384). Blodgett et al. (2021) find that many NLP fairness benchmarks are not meaningful measurement tools, while Subramonian et al. (2023) consider how differing task conceptualizations lead to benchmark disagreement. Denton et al. (2021) examine the implicit assumptions behind ImageNet, in particular the way its creators operationalize a particular conception of vision. Benchmarks considered here may encode notions of spuriousness whose appropriateness is context-sensitive.

**Practical applicability.** When simplifying real-world problems into benchmarks, we sometimes lose sight of our original intent (Selbst et al., 2019). Our results call for consideration of whether solving a benchmark corresponds to solving a real problem. For example, we observe that Dollar Street has a low task difficulty due to spurious correlation (Figure 5c). Thus, if our motive were to reduce computer vision's Western bias (Richards et al., 2023), one could ask whether developing methods optimized for spurious correlations is a helpful endeavor.

**Broader concerns.** Beyond validity, we might also ask about benchmark *acceptability*. One benchmark, CelebA (Liu et al., 2015), involves an association between hair color and a binary "gender" attribute (according to an external annotator), reinforcing views about gender that could be harmful to non-binary and gender nonconforming people (Keyes, 2018; Denton et al., 2020). It is our firm view that CelebA should not be used for benchmarking.

**Limitations.** One limitation of our work is that both the ERM Failure and Discriminative Power desiderata depend on well-defined group attributes. Similarly, while Convergent Validity can in principal be applied to any benchmark, our use of $K$ also requires group information for the reweighted model. More broadly, our evaluation only only considers benchmarks with well-defined group attributes, which may neglect realistic scenarios where distribution shifts are unknown. Exploration of spurious correlations benchmarks without attributes remains an exciting area for research.

The need for attribute information is also suggestive of a more subtle limitation. It is important to draw a distinction between satisfying a desideratum itself, and passing our test as currently implemented. For example, our test for Discriminative Power relies on standard deviation, which can be influenced by the presence of outliers. Whichever statistic used, reasoning about validity necessitates carefully assessing multiple streams of evidence, such as considering Discriminative Power in combination with other desiderata.

Our explicitly model-centric approach accounts for the relative utility of a spurious correlation to a specific model: as we note earlier, a color-based group feature is of no use to a grayscale-CNN (Arjovsky et al., 2019). Accordingly, our benchmark evaluation is conditional upon on the specific model architectures we used for testing mitigations methods, ResNet-50 and BERT base (chosen for their continued popularity in spurious correlations research). As the research community continues to develop new model architectures, we expect continued re-evaluation of the utility of current benchmarks (particularly with respect to ERM Failure) is likely to be necessary.

**Closing remark.** Benchmarks form an essential component of the way machine learning evaluates methods and draws conclusions. In the domain of spurious correlations, our analysis suggests that not all of our benchmarks yield equally meaningful results. We hope to have shown that benchmark choice matters: it leads to different conclusions; different recommendations; and ultimately better or worse deployed models.

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

## A  META-ANALYSIS OF PUBLISHED DATA

In Figure 1 we analyze previously-published results from the SubpopBench benchmarking library (Yang et al., 2023). We report similar findings analyzing the previously published results from the WILDS benchmarking library (Koh et al., 2021) in Figure 8. We take published worst-group test accuracies, as reported in these two papers, and compute the correlation (Pearson's $r$) between benchmarks. For SubpopBench, we use the "Worst Acc." columns from tables reported in Yang et al. (2023, Appendix E.1). For WILDS, we use the results reported in Koh et al. (2021, table 2).

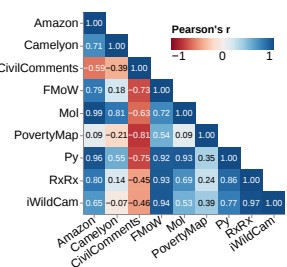

Figure 8: Correlation (Pearson's $r$) between worst-group accuracies across benchmarks in WILDS Koh et al. (2021).

## B  MUTUAL INFORMATION

Mutual Information (MI) between attributes and target labels is a common metric for evaluating the strength of a spurious correlation. Given a group-annotated dataset $(X, Y, A)$ where $X$ is the input data, $Y$ are the target labels and $A$ are group annotations, mutual information $I(Y; A)$ is defined as

$$I(Y; A) = \sum_{y,a} P(y,a) \log \frac{P(y,a)}{P(y)P(a)}. \tag{1}$$

However, mutual information will fail to account for the relative difficulty of learning to predict the target, or vice-versa learning to use the spurious correlation. As an illustrative example, consider a version of Waterbirds (Sagawa et al., 2020a) where the background is 100% noise. Whatever the mutual information between attributes and labels, this would not be a practically important spurious correlation. Conversely, in a dataset where the image of the bird is 100% noise, the task would be so difficult as to increase the reliance on the spurious correlation. We demonstrate this, showing the effect of applying background and foreground noise in Figure 9. Motivating our choice of a model-dependent statistic describing the task difficulty due to spurious correlation, mutual information is unable to account for the effects of foreground and background noise.

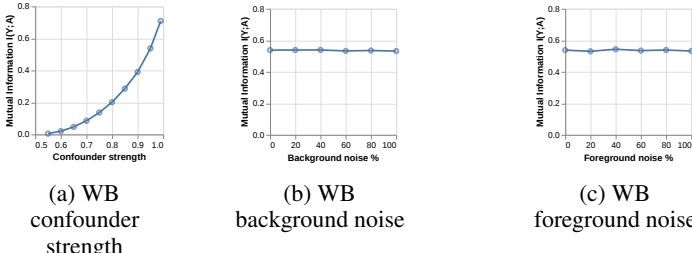

|     |     |     |
| --- | --- | --- |
| (a) WB confounder strength | (b) WB background noise | (c) WB foreground noise |

Figure 9: Mutual information $I(Y; A)$ for datasets with various synthetic modifications. By definition, increasing label-attribute correlation increases mutual information (a, b), but cannot account for the effect of background noise (c, d), foreground noise (e) or attribute noise (f). **Model-independent, data-only metrics are unable to capture the key factors driving task difficulty due to spurious correlation.**

## C  SUBPOPBENCH EVALUATION DETAILS

To evaluate the validity of common benchmarks, we build upon the comprehensive benchmarking library, SubpopBench, of Yang et al. (2023). The worst-group test accuracies we report follow the exact methodology specified by Yang et al. (2023). Reported results are the worst-group test accuracy on the test set. Group attributes are available at all times. Following Yang et al. (2023), we search 16 random hyperparameter configurations, and train 3 random seeds of the best performing configuration, where the "best" is the model with the highest worst-group test accuracy on the validation set. Vision benchmarks used a ResNet-50 model (He et al., 2016), pretrained on ImageNet-1k (Russakovsky et al., 2015). Language benchmarks used a BERT base uncased model (Devlin et al., 2019). For full details see Yang et al. (2023).

## D  BENCHMARKS

- Waterbirds (Sagawa et al., 2020a) is a binary image classification task of land-dwelling vs. water-dwelling birds, where the spuriously correlated attribute is water and land background scenes.

- CelebA (Liu et al., 2015) is a binary image classification task of blond vs. not-blond hair color, where the spuriously correlated attribute is annotator-perceived binary gender. **When training on CelebA, the only output models are capable of is a blond vs. not-blond binary judgment.**

- ImageNet Backgrounds Challenge, ImageNetBG (a.k.a. IN-9L Original; Xiao et al., 2021) is a multiclass image classification task, where images are selected from 9 coarse classes of ImageNet (Deng et al., 2009). Xiao et al. intend that the background is correlated with the classes, but there is no attribute information available in the SubpopBench formulation. Following Yang et al. (2023), mitigation methods fall back to class labels in the absence of attribute information.

- MetaShift (Indoor/Outdoor Cat vs. Dog; Liang & Zou, 2022) is a binary image classification task where the spuriously correlated attribute is indoor or outdoor scenes.

- NICO++ (Zhang et al., 2023) is a multiclass image classification task over common objects, plants and animals where the spuriously correlated attribute is one of (autumn, dim, grass, outdoor, rock, water).

- CheXpert (Irvin et al., 2019) is a medical image classification task over chest x-rays, with a binary classification into "finding" or "no finding". The spurious correlated attributes are patient race and gender, following (Yang et al., 2023).

- CivilComments (Borkan et al., 2019) is a binary text classification task of internet comments to be classified as toxic / not toxic. The spuriously correlated attribute is one of 9 group identities that are the target of the toxicity.

- MultiNLI (Williams et al., 2018) is a natural language inference task, comprising sets of sentences where sentences can either entail, contradict or be neutral with one another. The spuriously correlated attribute is the presence of negation words.

- Dollar Street (Gaviria Rojas et al., 2022) is a geographically diverse collection of household images, which in our work we frame as a multiclass object classification task, where the attribute information is geographic region. Although the original Dollar Street more classes, we filter them to only include classes that are present in every region, following (Gaviria Rojas et al., 2022). **When training on Dollar Street, models are only capable of outputting one of 42 classes, none of which are related to people.**

### D.1  CITYBIRDS

Citybirds is a new variant of Waterbirds where the confounding background scenes are urban or rural environments. We generate Citybirds using the Waterbirds generation scripts of Sagawa et al. (2020a), modifying the choice of background images. Background images are drawn from the *Places365* dataset (López-Cifuentes et al., 2020), with urban backgrounds from the "street" and "downtown" classes, and rural backgrounds from the "farm" and "field/cultivated" classes. The number of samples per group is matched between Waterbirds and Citybirds.

## D.2 ANIMALS VS. PLANTS (AvP)

AvP is a new binary classification task over diverse images of animals and plants, drawn from two geographic regions, Asia and Europe. The target label is animal or plant, and the group attribute is Asia or Europe. We construct AvP by sampling images from the *GeoYFCC* dataset (Dubey et al., 2021) of natural images, limiting our sample to only include classes that are within the hierarchy of the "animal.n.01" and "plant.n.02" ImageNet hierarchy. We explicitly exclude all images containing people. To identify region as either Asia or Europe, we map the country information provided by GeoYFCC to continents using the *pycountry_convert* Python package. The number of samples per group exactly matches Waterbirds. We validated that geographic information is harder to extract than the animals vs. plants classification task by comparing two linear models, one to predict class membership and another to predict geographic region, over representations extracted from a ResNet-50 (He et al., 2016) pre-trained on ImageNet-1K (Russakovsky et al., 2015). As expected, geographic performance was at chance level compared to much stronger performance for the target task.

See Figure 10 for the number of samples per group in each dataset.

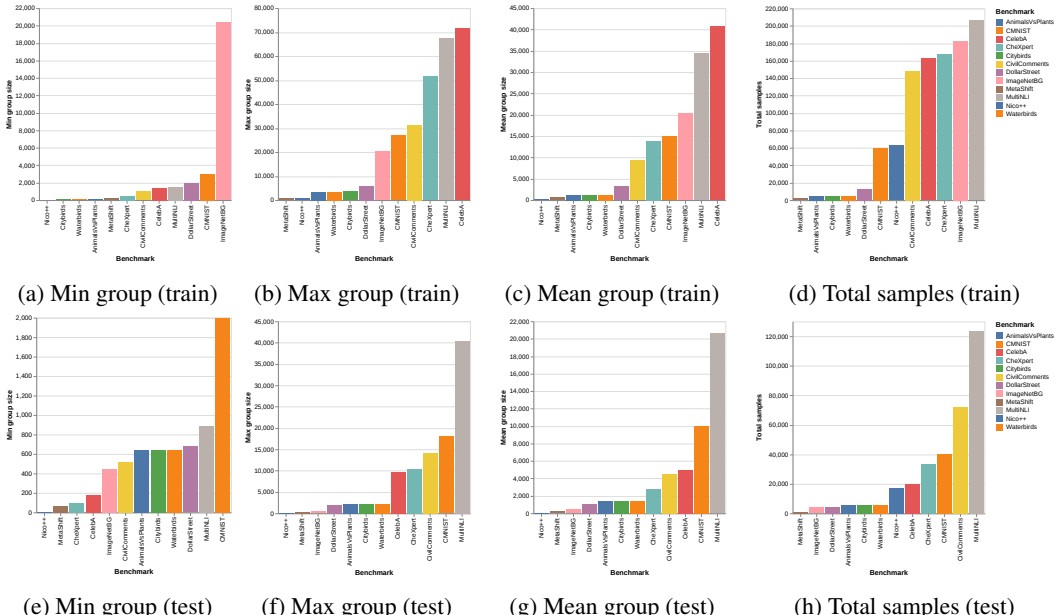

Figure 10: **(a)** Minimum group size per train dataset; **(b)** Maximum train group size; **(c)** Mean train group size; **(d)** Total number of train samples. **(e–h)** As above but for test set.

## E  $K$ VALIDATION BENCHMARKS

We implement a number of synthetic modifications to existing benchmarks in order to sanity check our measure $K$. To test for the effect of the amount of attribute–target correlation, we modify Waterbirds and CelebA. For Waterbirds, use Sagawa et al. (2020a)'s generation scripts, varying the "confounder strength" argument. For CelebA, we use subsampling to balance out the effect of the hair color and gender correlation, while keeping the total number of samples constant.

We also test version of Waterbirds with various degrees of RGB Gaussian noise applied to the background images and the foreground images, though we note that the outline of the bird (if not its detail) is still visible even under high noise. Given that noise can easily be memorized by contemporary neural networks, we also produce a version with a solid gray background, finding equivalent results to the 100% noise background case. Finally, we test for the effect of attribute noise by randomly flipping a proportion of the annotations, finding the $K$ becomes less able to detect the spurious correlation as attribute noise increases.

## F $K$ IMPLEMENTATION DETAILS

To calculate $K$, we train two models, $M_{ERM}$ and $M_{RW}$. $M_{ERM}$ is a base model (see Appendix C) trained to minimize cross-entropy loss using Adam (Kingma & Ba, 2017) with learning rate $1e-3$ until validation loss convergence. $M_{RW}$ is trained in the same way, though using a reweighted loss function, where the weight of each sample is proportional to the number of samples in each group.

Let $N^{\text{train}}$ be the number of samples in the training set, and $N^{\text{train}}_{a_i,y_i}$ be the number of samples where the attribute $a_i$ and target $y_i$ match the current sample $x_i$, then the weight for sample $x_i$ is

$$w_i = \frac{N^{\text{train}}}{N^{\text{train}}_{a_i,y_i}} \,. \tag{2}$$

Given that our approach is to compare a model using the spurious correlation with one penalized for doing so, it is possible to implement $K$ with an alternative reference method. We evaluate the robustness of our approach to choice of reference method by computing $K_{GroupDRO}$ as follows:

$$K_{GroupDRO} = \log \frac{P(Y^{\text{WG}}_{\text{test}}|X^{\text{WG}}_{\text{test}}, M_{GroupDRO})}{P(Y^{\text{WG}}_{\text{test}}|X^{\text{WG}}_{\text{test}}, M_{ERM})} \,,$$

where $M_{GroupDRO}$ is a model trained exactly as above but using GroupDRO (Sagawa et al., 2020a) instead of loss function reweighting. Figure 11 shows that $K_{GroupDRO}$ exhibits similar trends to those of $K_{RW}$ reported in Figure 3. In Figure 12, over all benchmarks and synthetic modifications considered in our paper, we see an almost perfect correlation between $K_{RW}$ and $K_{GroupDRO}$. Our results indicate that our measure of task difficulty due to spurious correlation is not senstive to choice of reference method.

A natural consequence of a model-dependent metric is a degree of sensitivity to the optimization procedure used when training the underlying model. For example, it is possible that $K$ may vary according to the hyperparameters chosen for training $M_{ERM}$ and $M_{RW}$. To evaluate this, we recalculate $K$ under different learning rates and batch sizes, for each of the benchmarks reported in Table 2, and evaluate whether the resulting sets of $K$ are consistent with $K$ as originally reported. We vary hyperparameters in lockstep between the two models, such that the learning rate for both models is always equal, as is the batch size, resulting in a like-versus-like, rather than best-versus-best comparison. Note that for the batch size experiments, we exclude NICO++ and Dollar Street due to training instability resulting from unsuitable batch sizes. In Figure 13, we present the correlation (Pearson's $r$) of the resulting $K$ versus the reference implementation, finding that regardless of hyperparameter choice, $K$ appears to produce a significantly highly positively correlated set of $K$ ($r > 0.9$; $p \leq 0.001$). Thus, we conclude that $K$ is practically robust in light of reasonable hyperparameter optimization.

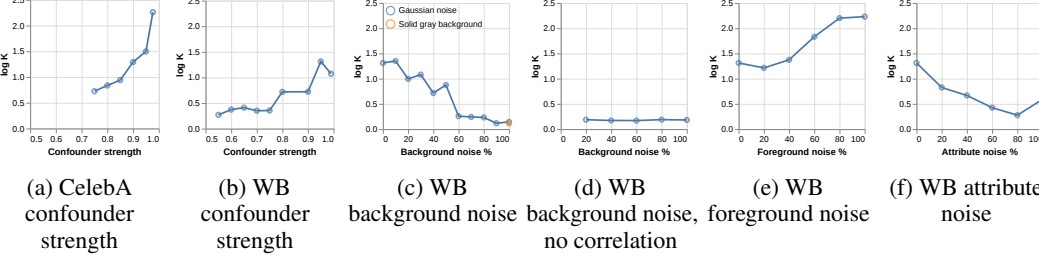

| (a) CelebA confounder strength | (b) WB confounder strength | (c) WB background noise | (d) WB background noise, no correlation | (e) WB foreground noise | (f) WB attribute noise |

Figure 11: Task difficulty due to spurious correlation, as measured by Bayes Factor $K$ computed using a gDRO reference rather than a reweighted loss reference, for datasets with various synthetic modifications. Increasing the label-attribute correlation (a, b) and foreground noise (e) increases $K$, while increasing background noise (c) or applying a solid gray background (c, orange point) decreases $K$, except in the case where there is no correlation (d). Attribute noise degrades the efficacy of $K$ (f). **Trends are consistent with Figure 3.**

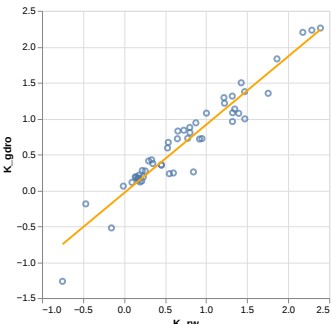

Figure 12: Recomputing Bayes Factor $K$ using a gDRO reference, rather than a reweighted loss reference, does not meaningfully change $K$. $K$ **is robust to choice of reference model.** (Pearson's $r$ = 0.97, $p < 1e - 6$).

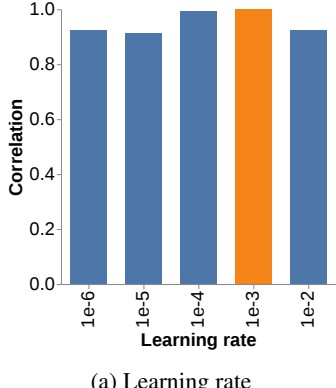

(a) Learning rate

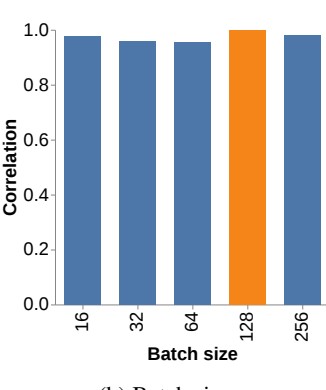

(b) Batch size

Figure 13: Correlation (Pearson's $r$) between $K$ as reported in main text (orange bar) and variants $K$ calculated with different **(a)** learning rates and **(b)** batch sizes, over benchmarks reported in Table 2. Training $M_{ERM}$ and $M_{RW}$ with other learning rates or batch sizes produces a $K$ that is consistently significantly highly positively correlated ($r > 0.9$; $p \leq 0.001$) with the original $K$ across all settings considered.

## G  EVALUATING METHOD SENSITIVITY

We test whether the variability in method performance is a function of varying $K$ by fitting a linear model using OLS to the worst-group test accuracies and the benchmark's $K$. We evaluate sensitivity in terms of the proportion of variance explained, using the linear model's Coefficient of Determination, $R^2$. A higher $R^2$ indicate that $K$ explains more of the variance in method performance, indicating greater sensitivity.

