# OpenReview forum: "Reassessing the Validity of Spurious Correlations Benchmarks"
_ICLR.cc/2025/Conference — Submitted to ICLR 2025_

### Official Review · Reviewer_fmNP · 2024-10-22

**Soundness:** 2
**Presentation:** 2
**Contribution:** 2
**Rating:** 3
**Confidence:** 4

**Summary:**

The paper investigates the problem of spurious correlations and the fact that results are inconsistent across benchmarks. It demonstrates that the top-performing methods on one benchmark may perform poorly on another, revealing significant benchmark disagreements. In particular, the authors show that some methods while achieving best on some benchmarks they perform among the bottom 3 on other benchmarks. To address this, the authors propose three desiderata for a valid benchmark: ERM Failure, Discriminative Power, and Convergent Validity. Their analysis shows that many benchmarks and mitigation methods fail to meet these criteria, questioning their effectiveness. The paper also provides guidance for selecting appropriate benchmarks based on specific tasks.

**Strengths:**

1. The paper's objectives and goals are clearly articulated.

2. The problem addressed is a longstanding challenge in machine learning, as defining spurious correlations and constructing relevant attributes is difficult. This paper delves deeply into the reasoning behind these challenges and explores the properties that datasets should possess to qualify for evaluating spurious correlations.

**Weaknesses:**

The paper seems to have been rushed for the submission. There are several errors, mistakes, typos, in addition to a comment left out by the authors regarding one of their figures that reads "Can we make x axis bigger? To hard to read even zooming" in line 242.

I will list below a non-exhaustive list:

1. Line 242 "Can we make x axis .. ".

2. Stay consistent "Figure" vs "fig" vs "Fig". It should always be capitalised "F" but at least stay consistent on the abbreviation or not.

3. Similarly to the above, also Appendix X, Figure Y, Table Z, Equation T all need to have the first letter capitalized.

4. Lines 202, 204 are missing extra spaces @ "Citybirdsshould" and "(AvP)has. There are a number of these

5. Figures are poorly presented. Figure 1 for instance, is hard to read (particularly figure 2). Make it bigger or change the presentation. The text is too small.

6. Caption of Figure 1 seems wrong. It reads "best method on Waterbirds (DFR) is the second worst on NICO++". DFR performs 19 (worst) on Waterbirds and second best on NICO++ according to Figure 2b.

7. "of its" > "to its" @ line 92.

8. Figure 4 is very poorly presented, xlabel, ylabels, and legends are all small.

9. Lots of white space in Figure 5. You can make it better. and enlarge the plots.


General weakness:

1. The paper focuses on image classification, which is to some extent an outdated setup and less exciting compared to newer domains.

2. All datasets require attributes, which is less realistic in real-world scenarios. The paper aims to provide a practical guide for practitioners deploying their models, but it is unlikely to encounter a test dataset where the attributes are known a priori.

3. The most interesting experiments are those presented in Table 2, as they provide evidence that filtering benchmarks based on the proposed desiderata helps in capturing and measuring spurious correlations. However, more experiments are needed. The details of this experiment section are poorly presented, and the rationale for selecting particular methods (GroupDRO and ReSample) is not adequately explained. It would also be valuable to investigate the same experiments using a different set of starting datasets to assess the impact of applying the desiderata on final performance.

**Questions:**

See above. I believe the paper is not yet ready for publication. The presentation is poor and the paper still needs to conduct a few experiments justifying the proposed desiderata in addition to better justify the experiments when attributes are actually required.

---

> ### Author Response · Authors · 2024-11-15
> **Thank you for your feedback**
>
> Thank you for your detailed and constructive feedback, and we apologize for giving you the impression that our submission had been rushed. We hope that we can assure you that this was not the case. We’ve just uploaded a new version addressing some of your feedback, including significantly larger figures. To respond to a few of your comments in turn:
>
> ### Re general weakness 1
> * The benchmarks we considered currently only cover classification settings, but they are not limited to computer vision: two NLP benchmarks, CivilComments and MultiNLI, are already included in our analysis.
> * More broadly, however, we agree that spurious correlations are likely to continue to be a problem in more general settings and newer domains, including both text and image generation. The set of benchmarks we analyze is intended to cover the majority of present research into mitigating spurious correlations.
> * We’d be happy to supplement our list if you have specific suggestions of existing benchmarks that we’ve neglected.
>
> ### Re general weakness 2
> * We discuss the reliance of our work, and all benchmarks within, on available (and high-quality) attributes on lines 521–530, and we’d be happy to expand this section if you think anything is missing.
> * We would like to stress that our “practical guide” in section 5 is intended for practitioners faced with a choice between competing methods for mitigating spurious correlations. In reality, a researcher without access to a group-annotated test or audit set is unlikely to be considering this suite of methods in the first place.
> * While an evaluation of benchmarks without group annotations would be a great next step, we don’t consider it within the scope of our contribution here.
>
> ### Re general weakness 3
> * If you have any specific recommendations for what you’d like to see, in terms of experiments, or to improve the presentation of section 5, we’d be more than happy to revise accordingly.
> * To help clarify our results, we compare three approaches in section 5 and table 2. (1) Picking the best method according to performance averaged over all benchmarks. This produces GroupDRO. (2) Picking the best method according to performance averaged over only valid benchmarks, according to section 3. This produces ReSample. (3) Picking the best method _on the closest benchmark according to K_. This produces a different method per test dataset, and often improves performance compared with (1) and (2), as noted on lines 477–481.
> * Table 2 currently covers 9 datasets, including Dollar Street to test the applicability of our approach to a held-out starting dataset, i.e., one not previously considered in our work. Do you have additional tests you’d like to see here?
>
> ### Re weakness 6
> * The caption for Figure 1 appears to be correct. In Figure 1, the legend on the color bar reads: “Rank (higher better)”. DFR is the best performing method on Waterbirds (rank 19), while the second worst on NICO++ (rank 2).
> * We have now clarified this in an updated caption.

---

> > ### Comment · Reviewer_fmNP · 2024-11-25
> > **Follow up**
> >
> > I have read the reviews from all reviewers and the submitted rebuttals by the authors.
> >
> > First, I would like to take this opportunity to thank the authors for their efforts in both the paper and the rebuttal. However, I am still concerned about the core of the experiments and the overall focus of the paper. The problem of classification feels somewhat outdated as a setup for studying spurious correlations. While the problem is of significant interest, I think it extends beyond the narrow scope of classification as presented in this paper. Moreover, even setting aside the fact that the work focuses on classification, the reliance on attributes appears vacuous in the era of large foundation models.
> >
> > I believe this is generally good work; however, further efforts are needed to address the two points mentioned above. Addressing these points would likely result in a fresh, entirely new paper, hence my decision.

---

> > > ### Author Response · Authors · 2024-11-25
> > > **Response to follow up**
> > >
> > > Thank you for responding to our review, and we're pleased you consider our work to be generally good. We also hope you found our updated revision to have addressed your concerns regarding layout and presentation.
> > >
> > > We'd like to take this opportunity to clarify why we disagree with the two points you make in your latest comment: 1) that the focus of classification is outdated, and 2) that reliance on group attributes is irrelevant in the age of large foundation models.
> > >
> > > A quick Google Scholar search suggests that 70+ papers published in 2024 rely on Waterbirds, which is just one of the benchmarks investigated in our evaluation. While LLMs bring new challenges, and exciting new considerations, evaluation on group-labelled classification datasets remains a cornerstone of the worst-group optimisation and group robustness community. It is highly unlikely that researchers developing new group-robust optimization algorithms, or new regularizers, will reach for large language models to begin their evaluations.
> > >
> > > The benchmarks we choose as a community---which are overwhelmingly drawn from those evaluated in our work---determine which methods we think are most promising, and therefore which methods get deployed in practice. As such, evaluating the validity of benchmarks that are currently used in active research remains of vital importance.

---

### Official Review · Reviewer_t9iK · 2024-10-24

**Soundness:** 3
**Presentation:** 2
**Contribution:** 2
**Rating:** 5
**Confidence:** 4

**Summary:**

The paper takes a critical look at a set of popular Spurious Correlation Benchmarks (SCBs), and shows that they often disagree with one another. The authors set out three desiderata that they think SCBs should exhibit based on the performance of different methods on the worse group accuracy. Specifically they claim a good SCB should exhibit a failure case of vanilla ERM, have strong Discriminative Power and Convergent Validity. The paper then evaluates how well these desiderata are satisfied by the set of SCB in question. The authors introduce a metric “K” to measure the difficulty of SCBs due to spurious correlations, After establishing a subset of the SCBs that satisfy the three desiderata, common domain generalisation approaches are assessed on this subset. Finally it is recommended practitioner also assess methods on these data sets or on data set similar in term of “K” to their data set of interest. In the Discussion section the authors discuss some weakness of their work and make some general recommendations for which SCB to use.

**Strengths:**

The papers main finding that Spurious Correlation Benchmarks (SCBs) often disagree with one another is interesting and definitely of interest.

The experiments performed are sound and presented in a clear and manner.

The prose of the paper are of good quality and in general it is easy to read and understand.

**Weaknesses:**

The biggest weakness of the paper is it is incorrectly titled, and the abstract is misleading. Spurious Correlations can be present outside of data sets with subpopulations shifts, however only subpopulation data sets and approaches have been considered in this work.  While the authors note this in the discussion section, I find this to still be insufficient. In its current state I think the work would be much better titled “REASSESSING THE VALIDITY OF SUBPOPULATION SHIFT BENCHMARKS”. With this title and a little rewriting to narrow the focus to these data sets and mitigation strategies I think the paper would be much better.

“Spurious Correlations” or “shortcuts” are typically defined as decision rules that perform well on standard benchmarks but fail to transfer to more challenging testing conditions, such as real-world scenarios (Geirhos R et al. 2020). This phenomena only requires a distribution shift between test and train environments. The link between group performance and spurious correlations is critically missing from the paper.

This has the following issues:
1)The assumption of having access to group information limits the usefulness of the desiderata to subpopulation shift benchmarks.
2) How data set are grouped into subpopulation would likely have a large impact on these desiderata, how robust the desiderata are to the merging or splitting of groups has not been explored. I would suspect that the desiderata would be very sensitive hence more detail here seems necessary.
3) There is no explanation of the different groups for the data sets in question, no detail on how the groups were selected. Or how to select useful groups when they have not be provided
4) Many (possibly all) of the mitigation strategies require group labels. Many Spurious Correlation mitigation strategies that don’t require group labels have not be considered, Feature Sieve, Deep feature reweighing, or the ensemble approach of (Teney et al 2022b) to name just a few.

All in all this paper just focuses on subpopulation shift benchmarks, hence the title and abstract and introduction should reflect that, and the effect of quality of the sub population labels should be explored.

The recipe for practitioners comparing mitigation methods on similar data sets in terms of K (Line 416-420), assumes access to data of the test domain to compute K. This requires the domain shift is known at train time.  This limits the usefulness of the approach as it assumes one has access to “clean” test data but insufficient to train on directly.

**Typos:**

Line 242: - author comment left in

Line 140: ANother

Line 102: correctionS - should be single

**Refs**

Geirhos R, Jacobsen JH, Michaelis C, Zemel R, Brendel W, Bethge M, Wichmann FA. Shortcut learning in deep neural networks. Nature Machine Intelligence. 2020 Nov;2(11):665-73.

Hermann KL, Mobahi H, Fel T, Mozer MC. On the foundations of shortcut learning. arXiv preprint arXiv:2310.16228. 2023 Oct 24.

Damien Teney, Maxime Peyrard, and Ehsan Abbasnejad. Predicting is not understanding: Recognizing and addressing underspecification in machine learning. In European Conference on Computer Vision, pp. 458–476. Springer, 2022b.

**Questions:**

**Questions**

What are the groups for the data set you consider?
How would extend your desiderata to setting where you did not have group labels?
How sensitive are your desiderata to the merging of groups or splitting of groups?


**Suggestions**

This paper focuses on subpopulation shift benchmarks, hence the title and abstract and introduction should reflect that, and the effect of quality of the sub population labels should be explored.

In its current state I think this work would be much better titled “REASSESSING THE VALIDITY OF SUBPOPULATION SHIFT BENCHMARKS”. With this title and a little rewriting to narrow the focus to these data sets and mitigation strategies I think the paper would be much better.

The 3.4.1 1. Is commonly referred to as “predictivity” and 2, and 3 are know as the “availability” Hermann et al. 2024 It’s also not clear to me what the different between 2 and 3 are.

---

> ### Author Response · Authors · 2024-11-21
>
> Thank you for your detailed review. We’ve uploaded a new version addressing the typos you describe, and have responded to your comments below. We’re open to making some changes to the framing as you suggest, and look forward to your feedback on our proposals below.
>
> ### Title, abstract, and framing
> * Thanks for suggesting this. We would be happy to consider adjusting the title and abstract, though we don’t fully agree that this paper needs to exclusively refer to subpopulation shifts instead of spurious correlations. We outline our reasoning below, and are open to working together to find a suitable middle ground here.
> * While the problems we’re investigating are sometimes referred to as subpopulation shift, this terminology is not consistently used in the literature. For example, as you point out in your review, Geirhos et al. (2020) refer to the problem as “shortcuts.” We feel that “spurious correlations” is immediately legible to the broader machine learning community, and as a result it’s the language we tend to use.
> * However, we propose restricting the focus of our abstract by modifying as follows (italics indicates new additions):
>   * “Neural networks can fail when the data contains spurious correlations, _i.e. associations in the training data that fail to generalize to new distributions._”
>   * “To understand this phenomenon, _often referred to as subpopulation shift or shortcut learning_, researchers have proposed numerous benchmarks upon which to evaluate mitigation methods.”
>   * “... we find that certain benchmarks are not meaningful measures of _how methods perform when correlations don’t generalize.”_
> * As for the title, we’d be open to modifying it to something that reflects the above proposals, though we’d first like to seek clarification from the other reviewers and the AC that this is a mutually acceptable change.
>
> ### Defining spurious correlations
> * Unfortunately, we don’t think the definition of “spurious correlation” is quite so clear cut, as we explain below.
> * Outside of machine learning, a statistician would define a spurious correlation as a correlation between two variables that aren’t causally related.
> * Within machine learning, while Geirhos et al. (2020, p. 665\) may define a shortcut as an association that fails to transfer to more challenging test conditions, Yang et al. (2023, p. 3\) define spurious correlations as those that are present in the training set but not in the test set. These aren’t quite the same thing.
> * Moreover, we don’t fully agree that a distribution shift needs to take place for a correlation to be considered spurious. For example, while CelebA is _commonly understood_ as exhibiting a spurious correlation between gender and hair color, this correlation is _equally_ present in both train and test sets. We think CelebA is an important benchmark to discuss in our work, but it doesn’t conform to a definition grounded in distribution shift.
> * Overall, setting out a precise and unifying definition of spurious correlation is not the ambition of our paper, nor an area we dedicate much time to.

---

> ### Author Response · Authors · 2024-11-21
>
> ### Group attributes
> * We agree that our work is limited to the setting where group attributes are available, and don’t try to suggest otherwise. We explicitly discuss this limitation in our discussion section, and would be happy to expand this if you feel there are specific things to call out here.
> * We would like to emphasize that our work isn’t to propose new benchmarks, but to _evaluate the validity of existing benchmarks_. Given that every benchmark we consider is a) equipped with group information, and b) commonly used for evaluating mitigations methods, we think this is a reasonable limitation.
> * Considering our three desiderata specifically—ERM failure, discriminative power, and convergent validity—they are explicitly intended to be applied to benchmarks where group attributes are available. For future benchmarks without group information, then we’d very likely want a different set of desiderata, at least one of which would need to cover some detectable failure at test time. We’d be happy to clarify this in section 3 if you think that would help.
> * We provide full details of each of the benchmarks, including how group attributes are defined, in Appendix D. With the exception of Dollar Street, where we define groups as geographic provenance, all group information is exactly as specified by Yang et al. (2023).
> * Our evaluation already includes methods that don’t rely on group attributes during training, such as Just Train Twice (JTT) and Learning from Failure (LFF).
>
> ### Predictivity
> * Thank you for the suggested terminology of predictivity and availability for section 3.4.1. We’re happy to update the paper to use these, and will cite Hermann et al. (2024) accordingly.
> * To clarify, the distinction between points (2) and (3) in section 3.4.1 refer to the availability of the group attribute, a, and the availability of the target label, y. Both are essential factors controlling task difficulty due spurious correlation.
>
>
> ### References
> Yang et al. (2023). Change is Hard: A Closer Look at Subpopulation Shift. ICML.
> Hermann et al. (2023). On the foundations of shortcut learning. arXiv:2310.16228.
> Geirhos et al. (2020). Shortcut learning in deep neural networks. NMI.

---

> > ### Comment · Reviewer_t9iK · 2024-11-21
> > **Group attributes**
> >
> > Thanks for your response. Just a quick reply as I fear a key part of my review might have not been clear enough (your response splits "Group attributes" and "Title, abstract, and framing" into two sections).
> >
> > For clarity my main objection is while you acknowledge "our<your> work is limited to the setting where group attributes are available" this isn't clear from a) the title b) the abstract c) the introduction. Spurious correlation's exist outside of settings where group attributes are available, hence these sections make it seem like your work is more general than you acknowledge it is.

---

> > > ### Author Response · Authors · 2024-11-21
> > >
> > > Ah! Thank you for the quick response and clarification, that is very helpful feedback.
> > >
> > > The point of our paper is to examine a set of existing benchmarks that are commonly-used for evaluating mitigations methods, and all of the benchmarks we examine are equipped with group attributes. However, we fully agree that the setting without group attributes is an important problem, that there are several attribute-free benchmarks for testing domain shift/OOD performance, and that we haven’t explored these in our work. In these cases the task is more to find generalizable functions, rather than mitigating failures for a specific slice of the data (as in the worst-group optimization literature). We'd definitely be happy to update the title, abstract, and introduction to reflect this distinction and the goal of our work more precisely.
> > >
> > > Perhaps a more appropriate title might be along the lines of “Reassessing the Validity of Group-Annotated Spurious Correlations Benchmarks”, or something to that effect?
> > >
> > > Overall, we hope you agree that the core of this paper---critically examining the validity of a set of benchmarks that are commonly used for evaluating mitigations methods---remains a sound and useful contribution (particularly given the significant body of work that still evaluates on these benchmarks), albeit one that is limited to the group-annotated benchmarks we evaluate.

---

> ### Comment · Reviewer_t9iK · 2024-11-22
>
> Yes, I agree within the scope of datasets and methods considered the core of the paper is sound and useful. I think it's important that the scope of the paper (which is somewhat limited) is correctly indicated from the outset.

---

> > ### Author Response · Authors · 2024-11-25
> >
> > Thank you again for your helpful suggestions. We've now updated the manuscript with an updated abstract and introduction to restrict our focus to group-annotated benchmarks, and expanded the discussion of this limitation in lines 523-525.
> >
> > We're happy to update the title, but will seek guidance on whether this is permitted before doing so. Please let us know if you would like to request any further changes.

---

### Official Review · Reviewer_qYSy · 2024-10-26

**Soundness:** 2
**Presentation:** 2
**Contribution:** 2
**Rating:** 5
**Confidence:** 3

**Summary:**

The paper assesses the quality of spurious correlation benchmarks and methods. The paper first develops three criteria desired for spurious correlation benchmarks and checks whether these are satisfied by some commonly used benchmarks. They then check which methods perform well across different benchmarks, and develop a new recommendation for choosing which method to use for a given dataset and model.

**Strengths:**

The results provide insights both about which benchmarks are good indicators of mitigating spurious correlations and which methods are robust across different benchmarks, which can be useful to a variety of practitioners.

**Weaknesses:**

1. Calculating K requires two full training runs (one with ERM, one with reweighting). This is extremely resource-intensive, and the empirical results do not seem to show a significant enough improvement to warrant such a cost.

2. The variety of spurious correlation benchmarks is a problem that has been addressed in previous work (Joshi et al., 2023; Yang et al., 2023). A more detailed comparison of the observations in this work versus those in previous work would be appreciated.

3. Some parts of the paper could be reorganized for clarity. A few specific points
- unnecessary comments on line 242
- lack of a dedicated related works section that puts the paper in the context of existing research (see previous comment)
- the discussion section jumps between many topics that are only loosely related to each other and the main paper, making it hard to follow

**Questions:**

I wonder how some datasets can have negative K (i.e. reweighting decreases performance)? This seems indicative of some confounding factors other than the identified spurious correlation, which may hinder the validity of the experimental results.

---

> ### Author Response · Authors · 2024-11-19
>
> Thank you for your helpful review! We’ve uploaded a new version incorporating your feedback, and respond to each in turn below.
>
> ### Re weakness 1
>
> * In section 5, we essentially make two proposals. First, we suggest using the closest benchmark to the target task. Second, we suggest K as a possible measure for establishing which benchmark is closest.
> * In our experiments using K to determine the closest benchmark, you are correct that this does require two full training runs. That said, we note that several mitigations methods already make use of multiple passes (e.g., JTT), yet are still widely-used by the community.
> * However, this might be a good moment to consider the practical alternatives. When practitioners need to choose a mitigations method, we imagine they might use one of three possible strategies:
>     1. Pick the method that performs best on average on previously-reported benchmarks.
>     2. Sweep over all methods, including all hyperparameters, and pick the best method on the dataset at hand.
>     3. Pick the method that performs best on the most relevant benchmark.
> * Of these, option (1) clearly requires the least compute, though is often outperformed by improved benchmark selection methods (see table 2).
> * Option (2) is naturally the most compute-intensive (i.e., sweeping a worst-case of 22 methods x necessary hyperparameters).
> * Our approach, option (3), falls somewhere in the middle. While requiring two full training runs, it is less intensive than the commonly-used strategy of sweeping over all methods.
> * Overall, while we agree our approach does come with a non-negligible compute cost, we consider it to be a reasonable compromise.
> * More broadly, we see our analysis in section 5 are more of a “first step” (line 460), and remain excited to see how future researchers might interpret the notion of benchmark similarity.
>
> ### Re weakness 2
> * Thanks for your suggestion to better articulate how our paper differs from prior work.
> * We discuss related work, including existing benchmarking efforts, in section 1.1, and explicitly discuss similarities with respect to other benchmarking efforts in lines 088–096. To respond specifically to your question about how our work differs from e.g., Joshi et al. (2023) and Yang et al. (2023), both works are concerned with benchmarking mitigations methods:
> * Joshi et al. (2023) construct 3 variants of a semi-synthetic benchmark, SpuCo, and use these to evaluate 9 spurious correlations mitigations methods. A key conclusion is that existing methods struggle with more complex spurious correlations, such as where the correlated attribute has high variance.
> * Yang et al. (2023) construct a benchmarking library, SubpopBench, allowing for the consistent evaluation of mitigations methods across several benchmarks. Yang et al. group benchmarkes into a proposed a taxonomy of subpopulation shifts.
> * In contrast, our work has a different aim: to evaluate the validity of the benchmarks themselves. This effort is distinct from, and complementary to, existing efforts to develop benchmarks and frameworks, and we hope that our desiderata can help guide the development of challenging benchmarks and their translation into real-world problems.
> * For example, our results suggest that, if using Yang et al.’s SubpopBench, one might consider including only the reduced set of valid benchmarks to ensure downstream utility.
> * Alternatively, our experiments with closest benchmark selection could prove useful in choosing between the several variants of the SpuCo benchmark proposed by Joshi et al., particularly for practitioners when faced with a new dataset.
> * Based on your feedback, we have updated the title of section 1.1 to be “Background and related work” and expanded our discussion of the focus of our work.
>
> ### Re weakness 3
> * Thank you for the comments regarding clarity. We have uploaded a new version addressing the errors you mention, reorganized the discussion to emphasize key themes, and expanded and retitled the background and related work section (see response to weakness 2). If you have further improvements you’d like to see, we’d be happy to implement them.
>
> ### Re question 1
> * Thank you for this important question.
> * As you pointed out, a negative K suggests that a simple ERM model outperforms a reweighted model. This would be the spurious correlation is not a principal factor in task difficulty (as we discuss on lines 373–376).
> * One alternative way of interpreting K is: “how far can simple reweighting get you?” For datasets where the spurious correlation is the dominant factor, reweighting should help. For datasets where other factors dominate, reweighting is likely ineffective, or even harmful overall (by reweighting uninformative samples, for example).
> * Note that we also evaluate an alternative implementation of K, asking the question “how far can gDRO get you?”, in appendix F, and find that both implementations produce almost identical results.

---

> > ### Comment · Reviewer_qYSy · 2024-11-21
> >
> > While JTT requires an extra pass, other spurious correlation mitigation methods require less overhead. The computation overhead of calculating K is not clearly expressed in the paper, and while the additional computational cost may be worth it if there is substantial performance improvement, the results in the paper only show marginal improvement (Table 2). I believe my point still stands.
> >
> > Thanks for the additional comparisons and clarifications.
> >
> > In summary, I believe the premise of the work---exploring the validity of spurious correlation benchmarks--is interesting, but the ideas are underdeveloped. Particularly the application of K to choose a mitigation method shows rather lackluster resultsin comparison to the computational cost. A deeper exploration of, for example, other uses for K, or **why** some benchmarks are better than others, would significantly strengthen the paper. While the existing work does provide some contributions, I believe my current score is still an accurate reflection of my evaluation.

---

### Official Review · Reviewer_BsWk · 2024-11-04

**Soundness:** 3
**Presentation:** 3
**Contribution:** 3
**Rating:** 8
**Confidence:** 4

**Summary:**

The paper investigates the validity and consistency of benchmarks used for evaluating methods that mitigate spurious correlations in machine learning models. Recognizing that current benchmarks often produce conflicting results—with certain methods performing well on one benchmark but poorly on others—the authors aim to understand the root of these disagreements. They propose three key desiderata for a valid spurious correlation benchmark: ERM (Empirical Risk Minimization) Failure, Discriminative Power, and Convergent Validity. To assess a benchmark’s validity, they introduce a model-dependent measure, the Bayes Factor (K), which quantifies task difficulty due to spurious correlation. Through an empirical study across multiple benchmarks, the paper identifies benchmarks that meet the proposed validity criteria and highlights methods that demonstrate robustness across varying benchmarks. Additionally, they offer practical recommendations for practitioners to choose benchmarks and methods tailored to their specific dataset characteristics, advocating for a systematic approach to benchmark selection in real-world applications.

**Strengths:**

- **Originality**: The paper presents a novel approach to evaluating spurious correlation benchmarks by proposing three validity criteria—ERM Failure, Discriminative Power, and Convergent Validity.
- **Quality**: The study is well-executed, with a thorough empirical analysis to assess the proposed validity criteria. The use of the Bayes Factor as a measure of task difficulty provides a quantifiable metric, helping to identify benchmark inconsistencies.
- **Clarity**: Definitions of key concepts, such as the three validity criteria, are well-explained. The practical recommendations provide actionable insights for researchers and practitioners selecting benchmarks.
- **Significance**: By focusing on the quality of benchmarks themselves, the paper addresses a critical gap in spurious correlation research. The findings could lead to improved benchmark selection practices, which are essential for evaluating and developing robust models across diverse domains.

**Weaknesses:**

The methods discussed in the paper currently omit some recent state-of-the-art algorithms and techniques in spurious correlation research published before July 1, 2024, which would strengthen both the related work and Section 4. For instance,
- Wang et al. "On the Effect of Key Factors in Spurious Correlation." AISTATS 2024.
- Yang et al. "Identifying Spurious Biases Early in Training through the Lens of Simplicity Bias." AISTATS 2024.
- Lin et al. "Spurious Feature Diversification Improves Out-of-distribution Generalization." ICLR 2024.
- Deng et al. "Robust Learning with Progressive Data Expansion Against Spurious Correlation." NeurIPS 2023.

Including these and potentially other relevant studies would make the paper more up-to-date. Even if not directly compared in Section 4, these works should at least be cited and discussed to reflect the current advancements in the field.

**Questions:**

1. There is an unresolved comment left in Line 242: “Can we make x axis bigger? Too hard to read even zooming.” This appears to have been unintentionally included in the submitted version and should be removed.
2. In lines 312-313, the paper states that $M_{ERM}$ is trained using ERM, while in lines 866-867, it is mentioned that $M_{ERM}$ is trained using the Adam optimizer. Could the authors confirm which training method was used and clarify any potential discrepancies?
3. Given the emphasis on benchmark selection, do the authors have insights into how the choice of model architecture might impact the validity of a benchmark? Are certain models more or less suitable for assessing spurious correlation benchmarks under the proposed criteria of ERM Failure, Discriminative Power, and Convergent Validity?

---

> ### Author Response · Authors · 2024-11-15
>
> Thank you for your considered and thoughtful review - we’re pleased you think that our paper addresses a critical gap, and that it could lead to improved benchmark selection practices. We’ve uploaded a new version addressing the typo you mention, and respond to your other comments below:
>
> ### Re weaknesses
> * Thank you for highlighting these papers. We are happy to add extended discussion of new advancements in the field that are not covered by the methods we evaluate, and we will explore the feasibility of adding these methods into section 4.
>
> ### Re question 2
> * You are correct that M_ERM was trained using Adam, but we don’t see these as conflicting. The model was trained to minimize the mean loss over the training samples (i.e., the empirical risk) using gradient-based optimization with an adaptive per-parameter learning rate (i.e., Adam). Whether using SGD, Adam, or RMSProp, as long as the gradient is w.r.t. the mean loss over the training set, we’d still consider this to be ERM. If you think this distinction needs further clarification, we’d be happy to update accordingly.
>
> ### Re question 3
> * Thank you for the wonderful question! The short answer is: yes, to a certain extent, we think model architecture should influence benchmark validity. To repeat the trivial example we use in the paper, if we were to choose a single-channel, grayscale, CNN architecture, then benchmarks where color is the spurious feature would be fairly useless.
> * More broadly, a core motivation in introducing our measure of task difficulty due to spurious correlation, K, is the idea that the strength of the spurious correlation is necessarily dependent on the model perceiving the correlation. That said, _in practice_, we imagine that many large image classification models would produce similar K for each benchmark—as a result of similar training data, and only small differences in architecture—and would be unlikely to substantially change our conclusions re benchmark suitability.
> * We chose the two model architectures considered, ResNet-50 and BERT base, as they remain the most frequent models considered in spurious correlations research, forming the typical basis upon which conclusions are drawn as to which methods perform best. As the research community continues to develop new models, repeatedly evaluating the utility of current benchmarks (particularly w.r.t. ERM Failure) is likely to be necessary.
> * We think this might make a nice addition to our discussion section, and thank you again for raising this question!

---

> > ### Comment · Reviewer_BsWk · 2024-11-27
> >
> > Thank you for your thoughtful responses and for addressing the issues raised in my review.
> >
> > **Re Weaknesses:**  I appreciate your acknowledgment of the missing citations and your willingness to extend the discussion of recent advancements in the field. However, I believe these additions should be reflected directly in the rebuttal revision, rather than deferred to future updates. Incorporating these state-of-the-art methods will provide a clearer and more complete picture of how your work fits within the broader research landscape.
> >
> > **Re Question 2:** Apologies for the confusion in my original question. I mistakenly referred to ERM when I meant to ask about the use of SGD. Specifically, the baseline papers, including (Yang et al, 2023), use the AdamW optimizer for all text datasets and SGD with momentum for all image datasets. Could you clarify whether the optimizer choice for $M_{ERM}$ aligns with these baselines in their respective settings? If not, do you think this difference in optimizer choice could have influenced the comparative results?
> >
> > **Re Question 3:** I appreciate your detailed response regarding the influence of model architecture on benchmark validity. Expanding on this point in the discussion section, as you suggested, would further enhance the manuscript by providing important context about how evolving architectures might necessitate re-evaluating benchmark utility.
> >
> > Thank you for the thoughtful clarifications provided in your responses. It is important that the proposed changes—such as the discussion of recent advancements and additional clarifications—are reflected directly in the rebuttal revision. Seeing these updates in the revised version would allow me to confidently maintain my positive assessment of the paper.

---

> > > ### Author Response · Authors · 2024-11-27
> > > **New revision and response to your comments**
> > >
> > > Thank you so much for coming back to us with this response and for your ongoing commitment to our paper.
> > >
> > > **New rebuttal revision:** We've just uploaded a new version with the following changes:
> > >
> > > 1. We now discuss the four papers you requested in our Background & Related Work section on lines 088--091. We've focused this on the theoretical contributions of each of these works, rather than the mitigations methods they propose, to remain inkeeping with the rest of the section. (Though see below re weaknesses.)
> > >
> > > 2. We've added a new paragraph to our Limitations section discussing the influence of model architecture and the expectation that these kind of analyses be repeated in the future (lines 527---532).
> > >
> > > We note that the deadline for uploading new revisions is today at noon UTC: we're of course happy to discuss further updates
> > > you'd like to see, but we won't be able to practically implement them as a result of the deadline.
> > >
> > > **Re weaknesses:** While we've updated the related work to discuss the additional references, we have not yet implemented all four additional methods in order to add them into section 4. We think this is a great suggestion to keep the paper fresh, and are committed to adding these methods, but it will take some time to both implement each method and run the full suite of evaluations against all of the benchmarks.
> > >
> > > **Re question 2:** Thank you again for this helpful clarification: you raise a great point. Our implementation of $K$ consistently uses an Adam optimizer for $M_{ERM}$. As long as it's Adam vs. Adam for _both_ $M_{ERM}$ and $M_{RW}$, we would be very surprised if the specific optimizer choice were to be of practical consequence here. This is particularly the case in light of Figure 12 in the appendix, where we show that replacing $M_{RW}$ with $M_{gDRO}$ results in almost identical values of $K$. We'd expect a similarly high correlation comparing an Adam-based with an SGD-based $K$. If you disagree and think we need an ablation, please do let us know.
> > >
> > > Thanks once again for the helpful suggestions and fruitful discussion!

---

### Comment · Area_Chair_sBXA · 2024-11-21
**Discussion Phase**

Dear Authors and Reviewers,

Thank you all for your contributions so far. As this paper is in a borderline situation with mixed scores, it’s important that we have a thorough discussion to reach a fair decision.

**To the authors:** It appears that Reviewer t9iK's comments were not addressed in your response—this may have been an oversight. Please provide a reply to their feedback at your earliest convenience.

**To the reviewers:** Please review the authors' replies and engage in further discussion if needed. If any concerns remain unresolved, feel free to seek further clarification from the authors.

We have until November 26th for this discussion. Your input is essential to making an informed decision.

Thank you all for your hard work and cooperation.

Best regards,
AC

---

> ### Comment · Area_Chair_sBXA · 2024-11-23
>
> Thank you Reviewers qYSy and t9iK for engaging in the discussion.
>
> Please, Reviewers BsWk and fmNP read the authors' replies and feedback from other reviewers. If any concerns remain, request clarifications. This is your last chance to engage.
>
> Thank you for your efforts.
>
> Best regards,
> Area Chair

---

### Author Response · Authors · 2024-11-25
**Summary of contributions**

We are grateful to all of the reviewers for their constructive reviews and discussion. We are pleased to see that:

* Reviewer BsWK found our “novel approach” to be “well-executed” and backed by a “thorough empirical analysis”, providing “actionable insights” and addressing a “critical gap.”
* Reviewer qYSy suggests that our insights “can be useful to a variety of practitioners.”
* Reviewer t9ik considers our finding that spurious correlations benchmarks disagree to be “definitely of interest”, our experiments to be “sound and presented in a clear manner”, and that the paper is “easy to read and understand.”
* Reviewer fmNP finds that we tackle a “longstanding challenge in machine learning”, with objectives and goals that are “clearly articulated.”

We’ve responded in detail to each reviewer below, and uploaded a new version that addresses several pieces of feedback, but would like to respond more broadly in order to highlight our core contributions, and indeed our non-contributions, to recap why we believe our work can serve as a useful resource for the community.

### Contributions

* We highlight that several commonly-used benchmarks used for evaluating spurious correlations mitigations methods exhibit strong disagreement when ranking methods.
* We re-evaluate the validity of eight benchmarks that have been, and continue to be, frequently used in the spurious correlations, worst-group optimization and group robustness literature. Given that many researchers in 2024 continue to benchmark their methods on these datasets, we think a critical examination is essential.
* To do so, we introduce three properties that specify the minimum bar which group-annotated spurious correlations benchmarks should pass to be deemed valid, and find that three of eight benchmarks fail at least one test.
* We propose a simple measure, K, in an attempt to quantify the task difficulty due to spurious correlation, while accounting for both the availability/learnability of group and target labels and the strength of their association.
* We suggest that *using some method* to select a similar benchmark, rather than averaging over all benchmarks, can be of use to practitioners. We instantiate this with a case study, using K, as *one possible method*.

### Non-contributions

* We do not suggest that the set of benchmarks we include is exhaustive, and admit that it is limited to benchmarks with group attributes. Still, given that these benchmarks remain in frequent use today—even in the era of LLMs—we believe this to be an important endeavour. In particular, we expect that researchers developing novel algorithms for group-robust optimization will continue to evaluate on the benchmarks we consider, even if those algorithms are eventually applied in an LLM setting.
* We do not propose that K, a measure of task difficulty due to spurious correlation, is compute-efficient. The most important idea in section 5 is that of using the *closest* benchmark, and we hope that future researchers will identify stronger and more efficient methods for measuring benchmark distance.

Thank you once again to all reviewers for your time and effort.

---

### Meta-Review · Area_Chair_sBXA · 2024-12-13

**Metareview:**

The Authors investigate the role of benchmarks in evaluating mitigation methods for spurious correlations. In the first part, they demonstrate that commonly used benchmarks can yield poorly correlated, or even anti-correlated, results when applying the same mitigation strategies. They identify three key attributes that an effective benchmark should possess and support their claims through experimental results. Finally, they propose a metric to quantify the impact of spurious correlations on a dataset’s performance, recommending its use to select the most appropriate benchmark for a given application.

**Additional Comments On Reviewer Discussion:**

The Reviewers agree on the paper's significance and believe it could make a strong contribution once the ideas are further developed. However, the current version remains unpolished, with minor typos and more serious concerns outlined below.

The paper’s contributions can be summarized in two key points: 1. Critiquing the indiscriminate use of standardized benchmarks without considering the final application; 2. Proposing a novel benchmarking method. The Reviewers identified issues with both that need addressing before publication.

1. One critique is that the paper's scope isn’t clearly defined until Section 2. The paper initially addresses a broad issue—spurious correlations—before narrowing its focus to subpopulation shifts with known attributes. While the Authors acknowledge that there’s no consensus on definitions in the literature, I agree with the reviewers that more clarity is needed. I recommend the authors follow the suggestions from Reviewers t9iK and qYSy in a future submission.

2. Another major critique concerns the use of the proposed "K" metric. One concern is the additional computational cost it incurs. While the authors explain that the cost is inline with other strategies, they do not sufficiently address the cost of the evaluation itself. Furthermore, the authors should expand the "practical recommendations" section to include more mitigation strategies. The results shown are encouraging, but not enough to be convincing.

---

### Decision · Program_Chairs · 2025-01-22

Reject